# Chrysin Derivative CM1 and Exhibited Anti-Inflammatory Action by Upregulating Toll-Interacting Protein Expression in Lipopolysaccharide-Stimulated RAW264.7 Macrophage Cells

**DOI:** 10.3390/molecules26061532

**Published:** 2021-03-11

**Authors:** Eui-Baek Byun, Ha-Yeon Song, Woo Sik Kim, Jeong Moo Han, Ho Seong Seo, Woo Yong Park, Kwangwook Kim, Eui-Hong Byun

**Affiliations:** 1Advanced Radiation Technology Institute, Korea Atomic Energy Research Institute, Jeongeup 56212, Korea; ebbyun80@kaeri.re.kr (E.-B.B.); songhy@kaeri.re.kr (H.-Y.S.); jmhahn@kaeri.re.kr (J.M.H.); seohoseong@kaeri.re.kr (H.S.S.); 2Functional Biomaterial Research Center, Korea Research Institute of Bioscience and Biotechnology, Jeongeup 56212, Korea; kws6144@kribb.re.kr; 3Department of Pharmacology, College of Korean Medicine, Kyung Hee University, Seoul 02447, Korea; jjing0429@naver.com; 4Department of Food Science and Technology, Kongju National University, Yesan 32439, Korea; nxkkwxm@nate.com

**Keywords:** chrysin derivative, anti-inflammatory activity, toll-like receptor negative regulator, toll-interacting protein, macrophage

## Abstract

Although our previous study revealed that gamma-irradiated chrysin enhanced anti-inflammatory activity compared to intact chrysin, it remains unclear whether the chrysin derivative, CM1, produced by gamma irradiation, negatively regulates toll-like receptor (TLR) signaling. In this study, we investigated the molecular basis for the downregulation of TLR4 signal transduction by CM1 in macrophages. We initially determined the appropriate concentration of CM1 and found no cellular toxicity below 2 μg/mL. Upon stimulation with lipopolysaccharide (LPS), CM1 modulated LPS-stimulated inflammatory action by suppressing the release of proinflammatory mediators (cytokines TNF-α and IL-6) and nitric oxide (NO) and downregulated the mitogen-activated protein kinase (MAPK) and nuclear factor-κB (NF-κB) signaling pathways. Furthermore, CM1 markedly elevated the expression of the TLR negative regulator toll-interacting protein (Tollip) in dose- and time-dependent manners. LPS-induced expression of cell surface molecules (CD80, CD86, and MHC class I/II), proinflammatory cytokines (TNF-α and IL-6), COX-2, and iNOS-mediated NO were inhibited by CM1; these effects were prevented by the knockdown of Tollip expression. Additionally, CM1 did not affect the downregulation of LPS-induced expression of MAPKs and NF-κB signaling in Tollip-downregulated cells. These findings provide insight into effective therapeutic intervention of inflammatory disease by increasing the understanding of the negative regulation of TLR signaling induced by CM1.

## 1. Introduction

The inflammatory response is regarded as a series of complex and unbalanced processes involving multiple cells, and the innate immune system is a major contributor to early inflammation [1]. Ultimately, inflammatory processes are initiated by the activation of pattern recognition receptors (PRRs). Toll-like receptors (TLRs) are the best-characterized PRRs and recognize pathogen-associated molecular patterns (PAMPs) [2]. TLR activation in response to ligand binding induces the expression of various inflammatory mediators, resulting in a pathological inflammatory response [3]. TLR4 plays a key role in various diseases, such as inflammatory bowel disease, sepsis, Alzheimer’s disease, and cancers; it can induce prolonged and excessive inflammatory signals, resulting in serious damage to various tissues [4,5]. For example, TLR4 signaling activation triggers a cascade of events leading to inflammatory response of innate immune cells via NF-B activation as well as activated T cell responses, which is considered as the major immunological cause in the onset of inflammatory bowel disease [6,7]. Furthermore, TLR4 signaling activation promotes colitis-associated tumorigenesis by inhibiting enterocyte apoptosis in early tumor promotion [8,9]. Similarly, TLR4 activation is associated with development of Alzheimer’s disease. The upregulated TLR4 by aggregated Aβ leads to activation of microglia, as a principal innate immune cell in brain development, subsequently leading to systemic neuroinflammation via crosstalk with the complement and inflammasome [10,11]. Sustained inflammatory responses are major features of neurodegenerative disorders [12,13]. Therefore, pharmacological inhibitors of TLR4 are considered attractive drugs for the treatment of the above-mentioned diseases. 

Flavonoids are extensively present in various plants, and flavonoid-rich diets are thought to play a key role in the prevention and treatment of inflammatory chronic diseases [14]. In particular, many researchers have reported that various flavonoids have strong anti-inflammatory activities, which are related to their regulatory effect of TLR-related molecules [15,16,17]. For this reason, flavonoids and their derivatives have drawn wide attention as a therapeutic candidate for inflammatory disease [18,19,20]. Chrysin (C_15_H_10_O_4_) is one of the most commonly distributed flavonoids in vegetable and fruit plants, and has been widely studied for several decades owing to its physiological properties, including anti-inflammatory, antioxidative, anticancer, and antihypertensive effects [21,22]. In our previous study, we identified a new chrysin derivative (CM1) which exhibited greater anti-inflammatory activity in dendritic cells than chrysin and exerted protective effects in a mouse colitis model [23]. We had previously identified that CM1 is a hydroxyethyl derivative of chrysin that is produced by ionizing radiation from chrysin methanolic solution (Figure 1a) [24]. However, the mechanism underlying the anti-inflammatory effect of CM1-mediated negative regulation of TLR signaling has not yet been elucidated. Thus, we aimed to determine the molecular mechanisms of CM1, which could contribute to better understanding of therapeutic application for inflammatory diseases.

## 2. Results

### 2.1. Inhibitory Effect of CM1 on LPS-Induced Pro-Inflammatory Cytokine and NO Production Without Cytotoxicity

CM1 isolated from gamma-irradiated chrysin was used in all experiments. First, we examined the cytotoxicity of CM1 at various concentrations (0.5–2 μg/mL). Staurosporine was used as a positive control. As CM1 did not exhibit any toxicity toward RAW264.7 cells (Figure 1b,c), CM1 was used at concentrations <2 μg/mL in all subsequent experiments. Next, we examined whether CM1 played a modulatory role in the expression of lipopolysaccharide (LPS)-induced proinflammatory cytokines, such as TNF-α and IL-6. As shown in Figure 1d, cytokine (TNF-α and IL-6) and NO production in the culture supernatants of RAW264.7 cells was significantly increased following LPS treatment, but overexpression of the proinflammatory cytokines and NO was significantly inhibited in response to CM1 (1–2 μg/mL) treatment. These findings strongly suggested that CM1 plays a key role in modulating LPS-induced inflammatory activity by inhibiting proinflammatory cytokine and NO production.

### 2.2. Effect of CM1 on LPS-Induced NF-κB, MAPK Signaling and Tollip Protein

To further investigate the effect of CM1 on the expression of proinflammatory mediators, we measured the levels of MAPK and NF-κB signaling pathway intermediaries, because these pathways can regulate cytokine production in various immune cell types [25]. As shown in Figure 2a–c, activation of MAPK (phosphorylation of ERK and p38) and NF-κB (degradation/phosphorylation of IκB-α and nuclear translocation of p65) signaling was measured by immunoblotting. The results showed that CM1 significantly inhibited LPS-induced activation of MAPK and NF-κB signaling. Based on these and our previous results, we predicted that the mechanism of action of CM1 involved the modulation of LPS-induced proinflammatory cytokine expression. Several intracellular molecules, such as SOCS1, Tollip, and IRAK-M proteins, can regulate inflammatory action by negatively regulating TLR activity [26]. To determine whether the negative regulators of TLR mediate the anti-inflammatory effect of CM1, we examined the expression of Tollip, SOCS1, and IRAK-M in the background of CM1 treatment in RAW264.7 cells. As shown in Figure 2d–f, CM1 (2 μg/mL)-treated RAW264.7 cells—compared to untreated cells—showed a dose- and time-dependent increase in Tollip expression, but SOCS1 and IRAK-M expression remained unaffected. Additionally, immunoblotting revealed Tollip knockdown in the Tollip-shRNA‒transfected cells, but TLR4 expression did not differ between the Tollip shRNA-transfected and untransfected RAW264.7 cells in the background of CM1 treatment (Figure 2g). Based on these results, we hypothesized that the anti-inflammatory effect of CM1 was due to upregulation of Tollip.

### 2.3. CM1-Induced Inhibition of Pro-Inflammatory Mediator and Surface Molecule Expression via Tollip Upregulation 

To verify our hypothesis that CM1 suppressed macrophage maturation through Tollip, we quantified the expression of macrophage activation-related surface molecules, such as CD80, CD86, and MHC classes I/II. As shown in Figure 3, CM1 inhibited LPS-induced expression of costimulatory molecules (CD80 and CD86) and MHC class molecules (MHC-I and II); however, the levels of surface molecules in Tollip shRNA-treated cells—compared to those in the control LPS/CM1-treated cells—were recovered. To determine whether CM1-induced Tollip expression mediated the downregulation of proinflammatory mediators, such as cytokines (TNF-α and IL-6), NO, and COX-2, we stimulated Tollip-downregulated RAW264.7 macrophage cells using LPS. As shown in Figure 4a,b, we found that CM1-induced inhibition of NO, IL-6, and TNF-α production was prevented by Tollip silencing. Next, we confirmed the expression of iNOS and COX-2, as these two enzymes are thought to be the most important regulators of inflammatory progression [27,28]. As shown in Figure 4c, LPS-induced expression of iNOS and COX-2 was significantly inhibited by CM1. However, this inhibitory effect of CM1 was not observed in Tollip-downregulated cells.

### 2.4. CM1-Induced Inhibition of the MAPK and NF-κB Signaling Pathways via Tollip Upregulation

To determine whether CM1-induced Tollip expression was correlated with MAPK and NF-κB signaling, the phosphorylation levels of MAPKs (ERK, p38, and JNK), phosphorylation/degradation of IκB-α, and nuclear translocation of NF-κB p65 were analyzed in Tollip-downregulated and control RAW264.7 macrophage cells. As shown in Figure 5a,b, LPS-induced MAPK (ERK1/2, p38, and JNK) and IκB-α phosphorylation was inhibited by CM1 (2 μg/mL), but the inhibitory effect of CM1 (2 μg/mL) on LPS-induced MAPK phosphorylation was attenuated in Tollip-downregulated cells. In agreement with these results, CM1 significantly inhibited LPS-induced nuclear translocation of NF-κB p65 in control cells, but not in Tollip-downregulated cells. Collectively, these results suggested that CM1-induced Tollip expression is the key factor in the downregulation of LPS-induced MAPK and NF-κB signaling in macrophages.

## 3. Discussion

CM1—a new chrysin derivative produced using irradiation technology—exerts greater anti-inflammatory effect than the parent chrysin, and it is considered an attractive therapeutic candidate for inflammatory bowel disease [23]. In this study, we investigated the mechanism underlying the anti-inflammatory effect of CM1 that is known to act as an important regulator of the early phase of inflammation in activated macrophages. 

As demonstrated previously, TLRs are PRRs with key roles in innate immunity. Briefly, when various TLRs bind to specific ligands, antigen presenting cells (APCs), such as macrophages and dendritic cells, migrate to the lymph nodes, and activated APCs subsequently induce the production of high levels of proinflammatory cytokines, NO, and surface molecules [29]. LPS, a well-known PAMP found in the outer membrane of most gram-negative bacteria, is widely used as a TLR agonist to induce an inflammatory response in in vitro studies [30]. Here, we showed that CM1 inhibited the overexpression of proinflammatory cytokines, NO, and surface molecules in LPS-stimulated macrophages by inhibiting the activation of MAPK and NF-κB signaling pathways. 

SOCS1, Tollip, and IRAK-M have been reported to negatively regulate excessive TLR stimulation [26,31]. Among these proteins, Tollip is a critical negative regulator of TLR4-mediated proinflammatory signals and is considered an effective target for the inhibition of excessive cytokine production in inflammatory diseases and pathogenic infections [32]. In this respect, a previous study had revealed that Tollip was probably entirely or partly involved in the inhibition of MAPK and NF-κB signals [33]. Tollip inhibits IRAK-1 autophosphorylation by directly binding to IRAK-1 [34]. IRAKs are first signal transducers of TLR that importantly mediate activation of NF-κB and MAPKs [35]. In other words, Tollip activation inhibits inflammatory signal by impairing myeloid differentiation antigen 88 (MyD88)-dependent NF-κB activation and MAPKs pathways. In this regard, several reports have been demonstrated that Tollip-knockout mice display a more aggravated LPS-induced inflammatory response than normal mice [36]. Similarly, Tollip-knockout mice are more susceptible to dextran sodium salt (DSS)-induced colon injury [32]. On the other hand, several compounds that acts as inducer of Tollip have great anti-inflammatory activities [37,38]. In agreement with the above reports, we observed significantly high dose- and time-dependent expression of Tollip in response to CM1 treatment of macrophage cells. Tollip knockdown abolished the inhibitory effects of CM1 on LPS-induced overexpression of inflammatory mediators (iNOS-mediated NO, cytokines, COX, and cell surface molecules), and the MAPK and NF-κB signals were restored. These findings showed that CM1-induced Tollip expression—serving as a key regulator of the MAPK and NF-κB signaling pathways—regulated the expression of various genes during inflammatory responses in LPS-activated macrophages, thereby demonstrating the therapeutic importance of CM1. Our results provide better understanding of anti-inflammatory mechanism of CM1, which support its potential application for inflammatory disease therapy.

However, there are several limitations to the present study. Although our study was focused on Tollip function in macrophages, further mechanistic studies on other immunological diseases, such as allergic inflammation and autoimmune disease, are needed. Additionally, future studies using the Tollip-knockout mouse model as an inflammatory disease animal model should be performed. Specifically, we observed that CM1 had great protective efficacy against DSS-induced colitis in our previous study [23], however, it is yet unclear whether protective effects of CM1 are due to Tollip expression. Based on the present findings, further studies on whether inhibitory effects of CM1 in DSS-induced colitis are due to Tollip expression and these anti-inflammatory effects of CM1 are abolished in Tollip-knockout mice are recommended for in-depth understanding of mechanistic role of CM1 as a candidate for colitis therapy.

## 4. Materials and Methods

### 4.1. CM1 Preparation

CM1 isolated from gamma-irradiated chrysin was prepared in accordance with a previously described protocol [24]. Briefly, chrysin-methanol solution was irradiated at a dose of 50 kGy (10 kGy/h) using a cobalt-60 irradiator (point source AECL; IR-221; MDS Nordion International Co., Ltd., Ottawa, ON, Canada). After irradiation, a preparative HPLC 1260 Infinity System (Agilent Technologies, Santa Clara, CA, USA) was used to purify CM1. 

### 4.2. Reagents and Antibodies

Fluorescein isothiocyanate (FITC)-conjugated Annexin V/propidium iodide (PI) Kit, anti-CD80, anti-CD86, and phycoerythrin-conjugated anti-MHC-I and anti-MHC-II were purchased from BD Biosciences (San Diego, CA, USA). Lipopolysaccharide (LPS) from *Escherichia coli* O111:B4 was purchased from InvivoGen (San Diego, CA, USA). Anti-phosphorylated extracellular signal-regulated kinase (ERK)-1/2 monoclonal antibody (Ab), anti-phosphorylated p38 monoclonal Ab, anti-phosphorylated c-Jun N-terminal kinase (JNK) monoclonal Ab, anti-phosphorylated inhibitor of κB (IκB)-α monoclonal Ab, anti-IκB-α monoclonal Ab, anti-nuclear factor (NF)-κB (p65) polyclonal Ab, anti-lamin B polyclonal Ab, anti-Toll-interacting protein (Tollip) monoclonal Ab, anti-suppressor of cytokine signaling (SOCS) 1 polyclonal Ab, anti-interleukin (IL)-1 receptor-associated kinase (IRAK)-M monoclonal Ab, anti-TLR4 polyclonal Ab, anti-cyclooxygenase (COX)-2 polyclonal Ab, anti- inducible nitric oxide (NO) synthase (iNOS) polyclonal Ab, and anti-β-Actin monoclonal Ab were obtained from Cell Signaling Technology (Danvers, MA, USA). Goat anti-rabbit-IgG-horseradish peroxidase (HRP) Ab and goat anti-mouse-IgG-HRP Ab were purchased from Calbiochem (Merck KGaA, Darmstadt, Germany), and 3-(4,5-dimethylthiazol-2-yl)-2,5-diphenyl-2*H*-tetrazolium bromide (MTT) was purchased from Sigma-Aldrich (St. Louis, MO, USA).

### 4.3. Cell Culture

The murine macrophage cell line RAW264.7 was purchased from the Korean Cell Line Bank (Seoul, Korea). RAW264.7 cells were cultured in Dulbecco’s Modified Eagle’s Medium (Grand Island, NY, USA) supplemented with 10% fetal bovine serum (Gibco, Thermo Fisher Scientific Inc., Waltham, MA, USA), penicillin (100 U/mL), and streptomycin (100 U/mL) at 37 °C in an atmosphere of 5% CO_2_.

### 4.4. Measurement of Cytotoxicity

Cytotoxicity was assessed using the MTT assay with an Annexin V/PI Staining Kit (BD Biosciences), according to the manufacturer’s instructions. Cultured RAW264.7 cells (1 × 10^5^ cells/mL) were treated with CM1 in a 12-well plate and incubated for 24 h. The cell death pattern of RAW264.7 cells was analyzed by annexin V/PI analysis. The harvested RAW264.7 cells were washed with phosphate-buffered saline (PBS, pH 7.2) and stained with FITC-conjugated annexin V and PI. The cells were analyzed using FACSVerse^TM^ (BD Biosciences), and the results were analyzed using FlowJo software (Tree Star, Ashland, OR, USA). For the MTT assay, the culture medium was replaced with 1 mg/mL MTT (100 µL). The cells were incubated at 37 °C for 1 h, the medium was removed, and the formazan product was dissolved in dimethylsulfoxide. Absorbance was measured at 570 nm using a microplate reader (Zenyth 3100; Anthos Labtec Instruments GmbH, Cambridge, UK).

### 4.5. Measurement of NO Production

NO concentration in the culture supernatants was measured using the Griess method [39]. Griess reagent (100 µL, 1% sulfanilamide/0.1% N-(1-naphtyl-ethylendiamine in 5% phosphoric acid) was mixed with the supernatants (100 µL) obtained from the experimental RAW264.7 cultures. After 15 min, absorbance of the solution was measured at 540 nm using a microplate reader. Standard curve of freshly prepared NaNO_2_ (0–100 μM) was used to calculate nitrite concentration.

### 4.6. Measurement of Cytokines

Tumor necrosis factor (TNF)-α and IL-6 levels in the culture supernatants were determined using commercial Cytokine Detection ELISA Kits (BD Biosciences), as per the manufacturer’s instructions. Protein levels were determined by measuring absorbance at 450 nm using a microplate reader, as described in our previous report [40].

### 4.7. Generation of Tollip-Depleted Cells

Tollip-depleted cells were generated using a previously described method [41]. Briefly, short hairpin RNA (shRNA; Santa Cruz Biotechnology, Dallas, TX, USA) against Tollip consisted of a pool of 3–5 lentiviral vector plasmids each harboring target-specific 19–25 nucleotides (plus hairpin). Cells were cultured in the presence of these shRNA for 7 h to knockdown Tollip expression. Neomycin selection was employed to obtain stably transfected cells.

### 4.8. Immunoblotting

For extracting the cytosolic fraction, cells were lysed in PRO-PREP protein extract buffer (100 µL; iNtRon Biotechnology, Seongnam, Korea), containing 1 mM phenylmethanesulfonyl fluoride (PMSF), and protease inhibitor cocktail. The nuclear fraction was prepared by treating cells with cytosolic lysis buffer (100 µL; 10 mM HEPES (pH 7.9), 10 mM KCl, 0.1 mM EDTA, 0.5% Nonidet P-40, 1 mM dithiothreitol, and 0.5 mM PMSF) on ice for 10 min. Following centrifugation at 4000× *g* rpm for 5 min, the pellet was resuspended in nuclear extraction buffer (100 µL; 20 mM HEPES (pH 7.9), 400 mM NaCl, 1 mM EDTA, 1 mM dithiothreitol, and 1 mM PMSF) and incubated on ice for 30 min. The nuclear lysates were centrifuged at 16,000× *g* for 10 min, and the supernatant containing the nuclear extracts was collected and stored at −80 °C until use. Protein concentrations were determined by performing the Bradford assay, and proteins (10–30 μg) were resolved by SDS-PAGE, following which they were transferred onto a polyvinylidene difluoride membrane (Millipore, Burlington, MA, USA). The membranes were blocked in 5% bovine serum albumin (BSA) and then probed with the respective antibodies (Abs) for 4 h. The membranes were washed with Tris Buffered Saline containing 0.05% Tween 20 buffer and probed with HRP-conjugated secondary Abs at room temperature for 1 h. Protein bands, including those of mitogen-activated protein kinases (MAPKs) and NF-κB, were visualized using an ECL Advance Kit (Millipore). Western blot bands were quantified by image J. 

### 4.9. Analysis of Cell Surface Molecules

RAW264.7 cells were harvested and preincubated with 0.5% BSA diluted in PBS for 30 min. The cells were then probed with fluorescence-conjugated anti-CD80, anti-CD86, anti-MHC-I, and anti-MHC-II (BD Biosciences) antibodies at 4 °C for 30 min. All Abs were diluted 100-fold before use. After washing with PBS, the expression of the corresponding molecules was measured by flow cytometry. The data were analyzed using FlowJo software.

### 4.10. Statistical Analysis

All experiments were repeated at least thrice and showed consistent results. The significance levels in comparisons between samples were determined by one-way analysis of variance, followed by Tukey’s multiple comparison or unpaired Student’s *t* test. Statistical analysis was performed using GraphPad 5.0 (GraphPad Software, San Diego, CA, USA). The data in the graphs were expressed as the mean ± SD. Values of * *p* < 0.05, ** *p* < 0.01, and *** *p* < 0.001 were considered significant.

## 5. Conclusions

In this study, we found that CM1 effectively inhibited LPS-induced expression of proinflammatory mediators via the MAPK and NF-κB pathways, and that Tollip expression was a key regulator of the anti-inflammatory activity of CM1 in macrophages. These findings may provide an advanced understanding of the anti-inflammatory mechanism of CM1 and might aid the development of better therapeutic strategies for inflammatory diseases.

## Figures and Tables

**Figure 1 molecules-26-01532-f001:**
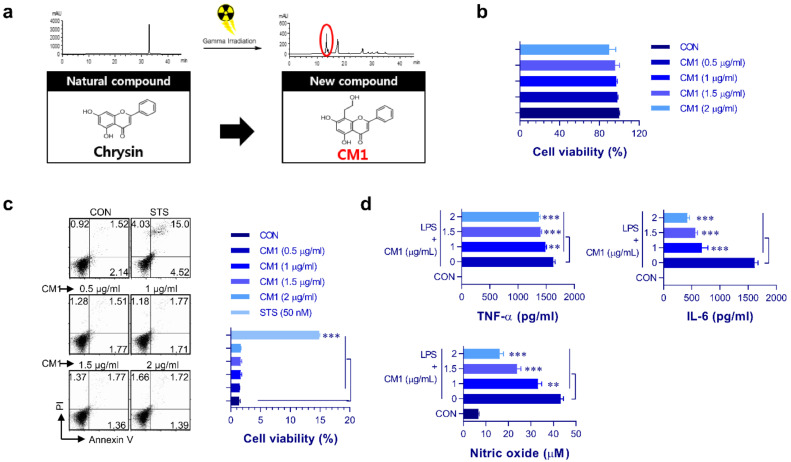
Cell viability and anti-inflammatory effect of CM1 on RAW264.7 cells. HPLC chromatogram and chemical structure of CM1 are shown (**a**). RAW264.7 cells were treated with CM1 (0.5, 1, 1.5, and 2 μg/mL) for 24 h. Cell viability was assessed using the MTT assay (**b**). RAW264.7 cells were treated with CM1 (0.5, 1, 1.5, and 2 μg/mL) and 50 nM staurosporine for 24 h, and cytotoxicity was assessed by annexin V/PI staining and flow cytometry (**c**). Staurosporine was used as a positive control for measuring cell death. RAW264.7 cells were treated with LPS (200 ng/mL) or LPS and CM1 (1, 1.5, and 2 μg/mL) for 24 h, and TNF-α, IL-6, and NO levels in the culture medium were analyzed by ELISA and Griess assay (**d**). All bar graphs show the mean ± SD of three samples from one representative plot among three independent experiments. Statistical analysis was performed by one-way ANOVA in conjunction with Tukey’s multiple test; ** *p* < 0.01 and *** *p* < 0.001. CON: untreated RAW264.7. Abbreviations: MTT, 3-(4,5-dimethylthiazol-2-yl)-2,5-diphenyl-2*H*-tetrazolium bromide; PI, propidium iodide; LPS, lipopolysaccharide; TNF, tumor necrosis factor; IL, interleukin; NO, nitric oxide.

**Figure 2 molecules-26-01532-f002:**
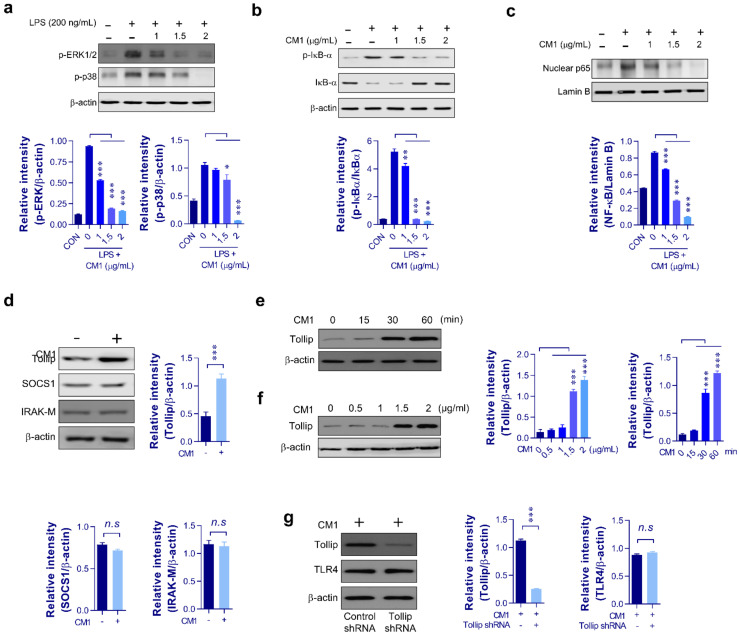
CM1-induced Tollip expression and inhibition of MAPK and NF-κB activation in RAW264.7 cells. RAW264.7 cells were pretreated with CM1 (1, 1.5, and 2 μg/mL) for 1 h and then stimulated with LPS (200 ng/mL) for 45 min (**a**–**c**). The cytosolic and nuclear fractions were subjected to SDS-PAGE, and immunoblotting was performed using specific Abs against p-ERK1/2, p-p38, p-JNK, p-IκB-α, IκB-α, and p65 NF-κB. β-Actin and lamin B were used as the loading controls for the cytosolic and nuclear fractions, respectively. RAW264.7 cells were treated with CM1 (2 μg/mL) for 1 h (**d**). Total cellular proteins were resolved by SDS-PAGE, and immunoblotting was performed using specific SOCS1, IRAK-M, and Tollip Abs. RAW264. 7 cells were treated with CM1 for the indicated time periods (**e**) or at indicated concentrations (**f**), and Tollip levels were measured by immunoblotting. RAW264.7 cells were transfected with the control and Tollip shRNA vectors (**g**). Cells were treated with CM1, and Tollip and TLR4 levels were measured by immunoblotting. One representative blot from three independent experiments is shown. Statistical analysis was performed by one-way ANOVA in conjunction with Tukey’s multiple test (**a**–**c**,**e**) or unpaired Student’s *t* test (**d**,**g**); *n.s*: no significance, * *p* < 0.05, ** *p* < 0.01, and *** *p* < 0.001. Abbreviations: MAPK, mitogen-activated protein kinase; NF, nuclear factor; Tollip, Toll-interacting protein; LPS, lipopolysaccharide; Abs, antibodies; p-ERK1/2, phosphor-extracellular signal-regulated kinase-1/2; p-p38, phosphor-p38; p-JNK; phosphor-c-Jun N-terminal kinase; IκB, inhibitor of κB; p-IκB; phosphor-IκB; SOCS1, anti-suppressor of cytokine signaling 1; IRAK-M, anti-interleukin-1 receptor-associated kinase; shRNA, short hairpin RNA; TLR4, Toll-like receptor-4.

**Figure 3 molecules-26-01532-f003:**
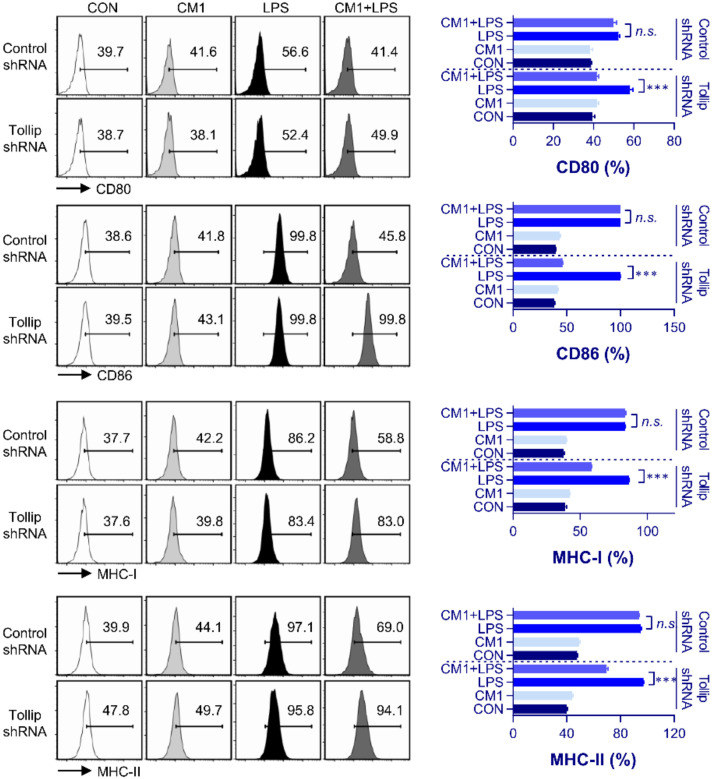
CM1-mediated suppression of macrophage surface marker expression through upregulation of Tollip expression. Control and Tollip shRNA-transfected RAW264.7 cells were pretreated with CM1 (2 μg/mL) for 1 h and then exposed to LPS (200 ng/mL) for 24 h. Cells were stained with anti-CD80, anti-CD86, anti-MHC class I, and anti-MHC class II. The percentage of positive cells is shown in each panel. Data are presented as the mean ± SD for triplicate determinations of one representative plot among three independent experiments. Statistical analysis was performed by one-way ANOVA in conjunction with Tukey’s multiple test; *n.s*: no significance and *** *p* < 0.001, versus LPS treatment alone. Abbreviations: Tollip, Toll-interacting protein; shRNA, short hairpin RNA; LPS, lipopolysaccharide.

**Figure 4 molecules-26-01532-f004:**
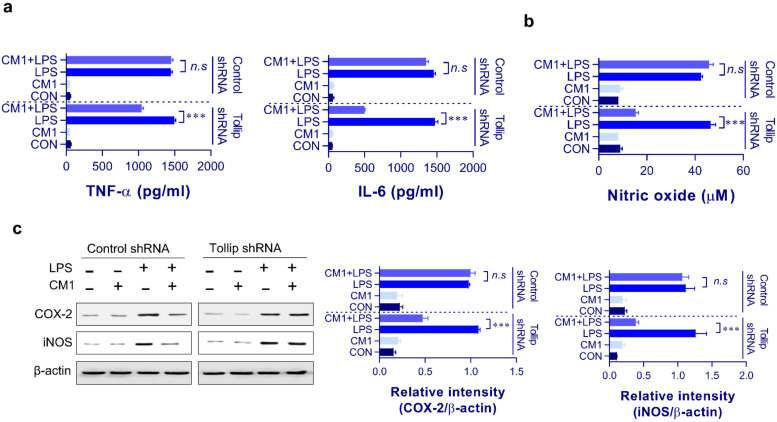
Tollip is indispensable for the inhibitory action of CM1 on proinflammatory mediators in LPS-treated RAW264.7 cells. Control and Tollip shRNA-transfected RAW264.7 cells were incubated with CM1 (2 μg/mL) for 1 h and then treated with LPS (200 ng/mL) for 24 h. TNF-α and IL-6 levels in the culture medium were measured by ELISA (**a**). RAW264.7 cells were pretreated with CM1 (2 μg/mL) for 1 h and then stimulated with LPS (200 ng/mL) for 45 min. NO level in the culture medium was measured using the Griess assay (**b**); *** *p* < 0.001. Data are presented as the mean ± SD for triplicate determinations of one representative plot among three independent experiments. Cytosolic fractions were analyzed to determine the COX-2 and iNOS protein levels using specific Abs (**c**). One representative blot from among three independent experiments is shown. Statistical analysis was performed by one-way ANOVA in conjunction with Tukey’s multiple test; *n.s*: no significance and *** *p* < 0.001. Abbreviations: Tollip, Toll-interacting protein; LPS, lipopolysaccharide; shRNA, short hairpin RNA; TNF, tumor necrosis factor; IL, interleukin; NO, nitric oxide; COX-2, cyclooxygenase-2; iNOS, inducible NO synthase; p-p38, phosphor-p38; p-JNK, phosphor-c-Jun N-terminal kinase; IκB, inhibitor of κB; p-IκB, phosphor-IκB; Abs, antibodies; NF, nuclear factor.

**Figure 5 molecules-26-01532-f005:**
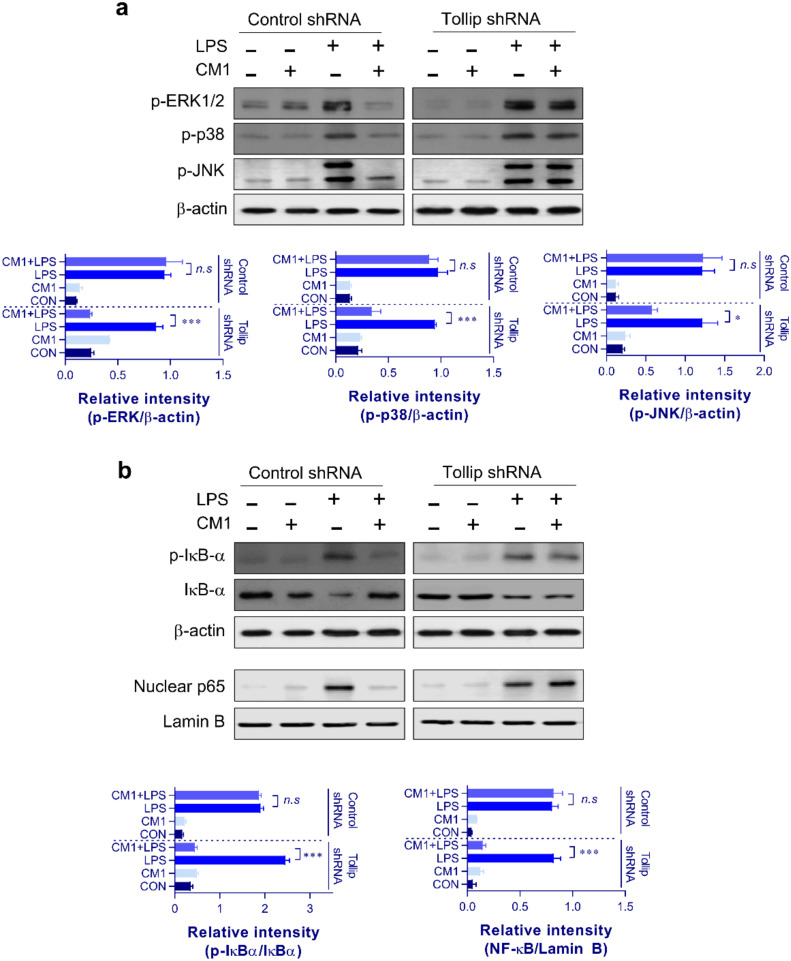
Tollip is indispensable for the inhibitory action of CM1 on MAPKs and NF-κB signaling in LPS-treated RAW264.7 cells. Control and Tollip shRNA-transfected RAW264.7 cells were incubated with CM1 (2 μg/mL) for 1 h and then treated with LPS (200 ng/mL) for 24 h. Control and Tollip shRNA-transfected RAW264.7 cells were pretreated with CM1 (2 μg/mL) for 1 h and then treated with LPS (200 ng/mL) for 45 min. Cytosolic fractions were analyzed to determine the p-ERK, p-p38, p-JNK, p-IκB-α, IκB-α, and β-actin protein levels using specific Abs (**a**). Nuclear fractions were analyzed to determine the p65 NF-κB and lamin B protein levels using specific Abs (**b**). One representative plot from among three independent experiments is shown. Statistical analysis was performed by one-way ANOVA in conjunction with Tukey’s multiple test; *n.s*: no significance, * *p* < 0.05, and *** *p* < 0.001. Abbreviations: Tollip, Toll-interacting protein; LPS, lipopolysaccharide; shRNA, short hairpin RNA; p-ERK1/2, phosphor-extracellular signal-regulated kinase-1/2, p-p38, phosphor-p38; p-JNK, phosphor-c-Jun N-terminal kinase; IκB, inhibitor of κB; p-IκB, phosphor-IκB; Abs, antibodies; NF, nuclear factor.

## Data Availability

Not applicable.

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
