# Peer review of "Chrysin Derivative CM1 and Exhibited Anti-Inflammatory Action by Upregulating Toll-Interacting Protein Expression in Lipopolysaccharide-Stimulated RAW264.7 Macrophage Cells"

_molecules, 2021, doi:10.3390/molecules26061532_

Round 1
Reviewer 1 Report
The manuscript under review entitled “Chrysin derivative, CM1, exhibit anti-inflammatory action by upregulating Toll-interacting protein expression in lipopolysaccharide-stimulated RAW264.7 macrophage cells” reports on an investigation about the mechanism underlying the anti-inflammatory effect of CM1, a new chirisin derivative, which seems to be an important regulator of the early phase of inflammation in activated macrophages. The Authors in this work evidenced how in macrophages CM1 was able to effectively inhibit LPS-induced expression of pro-inflammatory mediators via the MAPK and NF-kB pathways, and that Tollip expression seems to be a key regulator of this anti-inflammatory activity of CM1. This work will certainly be of great interest to the readers of this journal and, in my opinion, it can be considered for a possible publication after some minor revisions.
Best regards.
Comments for the Authors
- For all blots represented in the figures of this manuscript the Authors should please provide a densitomentric analysis to obtain a semiquantitative determination of the protein expression.
- The authors highlighted a marked decrease in some inflammatory markers to underline the anti-inflammatory action of CM1.
Have they thought of demonstrating whether this compound can also have an anti-inflammatory response modulation activity?
Have they thought about measuring some anti-inflammatory cytokines like il-10 or highlighting some suitable markers like Arg1?
In this regard they could add this type of data or argue on this aspect.
Author Response
The manuscript under review entitled “Chrysin derivative, CM1, exhibit anti-inflammatory action by upregulating Toll-interacting protein expression in lipopolysaccharide-stimulated RAW264.7 macrophage cells” reports on an investigation about the mechanism underlying the anti-inflammatory effect of CM1, a new chirisin derivative, which seems to be an important regulator of the early phase of inflammation in activated macrophages. The Authors in this work evidenced how in macrophages CM1 was able to effectively inhibit LPS-induced expression of pro-inflammatory mediators via the MAPK and NF-kB pathways, and that Tollip expression seems to be a key regulator of this anti-inflammatory activity of CM1. This work will certainly be of great interest to the readers of this journal and, in my opinion, it can be considered for a possible publication after some minor revisions.
Best regards.
- We appreciate the reviewer’s clear summary of our work and also thank your kind review. We have attempted our best to make equivalent corrections as your comments would indicate.
Comments for the Authors
Q: For all blots represented in the figures of this manuscript the Authors should please provide a densitomentric analysis to obtain a semiquantitative determination of the protein expression.
- The points that you have request for us to review is very reasonable to raise quality of this study. As you suggested, we added bar graphs that showed the relative intensity of each protein quantified using image J in figure 2, 4, and 5. All revised parts (Band quantification and statistical analysis) were marked with red (Line 134-135, 164-165, 175-176, 201-202, and 317).
Q: The authors highlighted a marked decrease in some inflammatory markers to underline the anti-inflammatory action of CM1. Have they thought of demonstrating whether this compound can also have an anti-inflammatory response modulation activity? Have they thought about measuring some anti-inflammatory cytokines like il-10 or highlighting some suitable markers like Arg1? In this regard they could add this type of data or argue on this aspect
- We appreciate reviewer’s excellent comments and completely agree with the comments. Actually, we have investigated that whether treatment of CM1 during dendritic cells (DCs) differentiation induce phenotypic change into tolerogenic state. Generally, tolerogenic DCs highly express anti-inflammatory molecules, such as IL-10, Arg1, TGF-β, indoleamine 2,3 dioxygenase 1 (IDO1), which strongly induce generation of regulatory T cells. Interestingly, we found that CM1 treatment during DC differentiation induces tolerogenic DCs characterized by high expression of IDO1. In this regard, the authors are carefully preparing our next publication and supplementary experiments.

Reviewer 2 Report
Overall an interesting, well-designed study providing insight into the potential effective therapeutic intervention of inflammatory disease by increasing the understanding of the negative regulation of 31 TLR signalling induced by CM1. The manuscript is well-written and formatted with sound use of figures. However, the authors need to address the limitations to the study in detail as opposed to simply stating there are several limitations. There are also other areas of the manuscript that require minor changes to further elaborate on key discussion points in order to strengthen the study findings as outlined below. My recommendation is that this study be accepted for publication with minor edits.
- Lines 44-47 where the authors discuss the key role TLR4 plays in various diseases, I recommend the authors include more relevant original articles to support this statement, the two references provided (Ref 4,5) only provide evidence for Alzheimer’s disease and inflammation/sepsis but not for cancer as an example. Furthermore, both of these references are review articles and while they do provide support for the statement made, original sources would strengthen this.
- Minor comment: I think that lines (69-72) should be discussed in the introduction rather than the results paragraph. This is important information to provide background to the study but does not necessarily fit the results paragraph. The first paragraph of results could simply begin with ‘CM1 isolated from gamma-irradiated chrysin was used in all experiments’.
- Given the focus of this manuscript is Tollip function in macrophages and the findings determine Tollip to be indispensable for the inhibitory action of CM1 on MAPKs and NF-κB signaling in LPS-treated 187 RAW264.7 cells, it would be beneficial for the authors to further elaborate either in the introduction or discussion (possibly expanding on Lines 212-215), on the findings of any previous relevant or related studies on Tollip function, also those that may have utilised the Tollip-knockout mouse model for related research to provide some further background as to where research related to Tollip is to date in the literature. This will also support the statement by the authors to utilise such models in future to expand upon the current studies findings (see next point).
- It would be beneficial for the authors to elaborate on specifically why it would be beneficial for future studies to use the Tollip-knockout mouse in the context of this studies findings. What would use of such a model achieve? How could this model provide further information for the anti-inflammatory mechanism 345 of CM1? (Lines 231-232).
- It is necessary for the authors to elaborate on the specific limitations of this study, that is outline each if the limitations rather than just stating there are several limitations (Line 228). This is particularly important for replication and design of future studies to overcome such limitations.
Author Response
Overall an interesting, well-designed study providing insight into the potential effective therapeutic intervention of inflammatory disease by increasing the understanding of the negative regulation of 31 TLR signalling induced by CM1. The manuscript is well-written and formatted with sound use of figures. However, the authors need to address the limitations to the study in detail as opposed to simply stating there are several limitations. There are also other areas of the manuscript that require minor changes to further elaborate on key discussion points in order to strengthen the study findings as outlined below. My recommendation is that this study be accepted for publication with minor edits.
- The authors would like to express a sincere appreciation to your kind review. We will try our best to make adequate corrections following your suggestions.
Lines 44-47 where the authors discuss the key role TLR4 plays in various diseases, I recommend the authors include more relevant original articles to support this statement, the two references provided (Ref 4,5) only provide evidence for Alzheimer’s disease and inflammation/sepsis but not for cancer as an example. Furthermore, both of these references are review articles and while they do provide support for the statement made, original sources would strengthen this.
- Thank you for your helpful comment. As you suggested, we described in detail the role of TLR4 in various disease with relevant references in introduction part (Line 47-56).
Minor comment: I think that lines (69-72) should be discussed in the introduction rather than the results paragraph. This is important information to provide background to the study but does not necessarily fit the results paragraph. The first paragraph of results could simply begin with ‘CM1 isolated from gamma-irradiated chrysin was used in all experiments’.
- Thank you for your reasonable comments. Following your suggestion, we revised sentences regarding background of study (figure 1a) in introduction.
Given the focus of this manuscript is Tollip function in macrophages and the findings determine Tollip to be indispensable for the inhibitory action of CM1 on MAPKs and NF-κB signaling in LPS-treated RAW264.7 cells, it would be beneficial for the authors to further elaborate either in the introduction or discussion (possibly expanding on Lines 212-215), on the findings of any previous relevant or related studies on Tollip function, also those that may have utilised the Tollip-knockout mouse model for related research to provide some further background as to where research related to Tollip is to date in the literature. This will also support the statement by the authors to utilise such models in future to expand upon the current studies findings (see next point).
It would be beneficial for the authors to elaborate on specifically why it would be beneficial for future studies to use the Tollip-knockout mouse in the context of this studies findings. What would use of such a model achieve? How could this model provide further information for the anti-inflammatory mechanism of CM1? (Lines 231-232).
It is necessary for the authors to elaborate on the specific limitations of this study, that is outline each if the limitations rather than just stating there are several limitations (Line 228). This is particularly important for replication and design of future studies to overcome such limitations.
- We totally agreed with your comments and would like to thank the reviewer for the professional comments. We added the mechanistic role of Tollip in MAPKs and NF-B regulation in discussion parts (Line 227-236).
- Furthermore, we additionally described the recommended design for further study in discussion section. Previously, we found that CM1 alleviated DSS-induced colitis symptoms, however, it is yet unclear that whether protective effects of CM1 in DSS colitis model are due to Tollip expression. Therefore, further studies that whether inhibitory effects of CM1 in DSS-induced colitis are due to Tollip expression and these anti-inflammatory effects are abolished in Tollip-knockout mice are needed. These information were added in Line 250-256.
